REGISTERED REPORT PROTOCOL

# Neural correlates of thematic role assignment for passives in Standard Indonesian

**Bernard A. J. Jap** *, **Yu-Yin Hsu**, **Stephen Politzer-Ahles**

Department of Chinese and Bilingual Studies, The Hong Kong Polytechnic University, Kowloon, Hong Kong

* bernard.jap@polyu.edu.hk

## Abstract

Previous studies of multiple languages have found processing differences between patient-first and agent-first word orders. However, the results are inconsistent as they do not identify a specific ERP component as a unique correlate of thematic role processing. Furthermore, these studies generally confound word order with frequency, as patient-first structures tend to be infrequent in the languages that have been investigated. There is evidence that frequency at the sentence level plays a significant role in language processing. To address this potential confounding variable, we will test a language where the non-canonical sentences are more frequent and are comparable to the canonical sentences, namely Standard Indonesian. In this language, there is evidence from acquisition, corpus, and clinical data indicates that the use of passive is frequent and salient. One instance of this difference can be demonstrated by the fact that it has been suggested that frequency may be the reason why Indonesian-speaking aphasic speakers do not have impairments in the comprehension of passives, whereas speakers of other languages with aphasia often do. In the present study, we will test 50 native speakers of Indonesian using 100 sentences (50 active and 50 passive sentences). If the neural correlates of thematic role processing are not observed in the critical region of the sentence (the prefix of the verb), this would suggest that the previous results were indeed influenced by frequency, but if we find that specific ERPs are connected to the hypothesized syntactic operations, this would further reinforce the existing evidence of the increased cognitive load required to process more syntactically complicated sentences.

This is a Registered Report and may have an associated publication; please check the article page on the journal site for any related articles.

## Introduction

For most individuals, understanding sentences is an effortless, uneventful process. However, early research has suggested that not all sentences are equal. Consider the passive structure: Compared to active sentences, passives are acquired and used later by children [1],take longer to process, and are prone to misinterpretation by adults [2]; they also pose difficulties for individuals with brain lesions [3].Debates regarding how humans process different types of sentences have evolved over time. The use of event-related potentials (ERPs) to study language

**Data Availability Statement:** All relevant data from this study will be made available upon study completion.

**Funding:** This research is funded by The Hong Kong Polytechnic University (UGC), Internal Research Fund (ID:A0038934). YYH, SPA, and BAJ are recipients of the grant. URL: https://www.polyu.edu.hk/rio/ The funders had and will not have a role in study design, data collection and analysis, decision to publish, or preparation of the manuscript.

**Competing interests:** The authors have declared that no competing interests exist.

processing is commonplace not only among language researchers, but also among psychologists and clinicians.

In recent years, the vast number of studies investigating neural responses during language processing have mainly focused on linguistic anomalies. The utilization of these violations has yielded distinct ERP responses. The P600, for example, is seen in response to syntactic violations such as violating the expected word order [4] or inflection [5], reflecting semantic integration [6],and the processing of long-distance wh-dependencies [7]. Other ERPs, such as the N400, appear as a result of anomalies such as semantic incongruence [8], the integration of the overall sentence meaning [9], or memory retrieval [6].

## Word order processing in ERP research

While the study of ERP responses to linguistic anomalies has been valuable, the tested structures are, by their own nature, anomalies which seldom appear in normal language use [10]. This leads to the question of whether the neural responses that are found in reaction to anomalies can also be observed during typical language comprehension. Not only is there less research on the topic of processing of typical sentences compared to studies using violation-paradigms in sentence processing, which might be considered surprising given that this is the type of language that individuals encounter in daily communication, but the few studies that exist have found little to no consistency and have been plagued by potentially confounding variables such as syntactic frequency. This subsection will discuss the studies that focused on word order effects in sentence processing.

Some studies have documented differences in the processing of different word orders, such as between subject and object relative clauses [10, 11], as well as of simple sentences such as the subject-object and the object-subject word order distinction in German [12]. Most of these studies have investigated languages with a canonical subject-first word order, either SOV or SVO, and have reached similar conclusions: Object-first structures require more effort to process than subject-first structures. For example, in the sentence with an object-embedded relative clause (1a), "The man" is not the agent of the first verb, thereby this does not adhere to the typical English word order in which the agent is placed in the first position [13]. This results in a predicted increase in syntactic processing demand (as reflected in reaction times or accuracy rates, see [2]). The sentence with a subject-embedded relative clause (1b), however, does not violate this expectation, as the NP1 of the sentence is the agent of the action.

1a) The man who the woman violently scolded, admitted the error.

1b) The man who violently scolded the woman, admitted the error.

Matzke et al. [12] compared object-before-subject (OS) to subject-before-object (SO) structures in German with the provision of case information through the use of articles. The temporary ambiguity of case information was also included as a factor by using feminine case markers on the article (*die*) of the NP1 which, in German, can signify both the accusative and the nominative case (2 and 3).

2) Object—Subject

Die begabte Sängerin entdeckte der talentierte Gitarrist.
*The gifted singer(Fem. Nom/Acc.) discovered the talented guitar player(Masc.Nom.).*
'The talented guitar player discovered the gifted singer.' [Ambiguous until 'der']

3) Subject—Object

Die begabte Sängerin entdeckte den talentierte Gitarrist.

*The gifted singer(Fem. Nom./Acc.) discovered the talented guitar player(Masc.Acc.)*.
'The gifted singer discovered the talented guitar player.' [Ambiguous until 'den']

In the masculine NP1 sentences, with regard to the Object-Subject compared to the Subject-Object structures in German, Matzke et al. [12] found a Left Anterior Negativity (LAN) for the critical time window (NP1) that continued for the rest of the sentence. Left fronto-temporal negativity was observed following the 2nd article in the Object-Subject condition. In the condition that was ambiguous up to the second article (as in examples 2 and 3), a P600 was found in the disambiguation section (the 2nd article 'der') of the Object-Subject compared to the Subject-Object structures. Matzke et al. [12] attributed the initial LAN on the NP1 to working memory. In a similar experiment, Schlesewsky, Bornkessel, and Frisch [14] found that the LAN was only observed in object-first non-pronominal NP1s in German, and not in pronominal NP1s (as shown in example 4).

4) Object-first pronominal structure

> Gestern hat <u>ihn</u> der Vater dem Sohn gegeben.
> Yesterday has it_ACC the_NOM father the_DAT son given
> 'Yesterday, the father had given it to the son'.

As such, the authors came to the conclusion that the LAN originates from a local syntactic mismatch via the violation of canonicity principles in non-pronominal NP1s, rather than from higher working memory usage due to dislocated objects in general.

Other studies which have investigated thematic role assignment processing through ERPs included Meltzer and Braun [10], who compared the processing difference between subject-embedded and object-embedded relative clauses in English, and they found a negativity at NP1 (400-800ms) and a positive shift at the offset of the relative clause for reversible clauses. Jackson, Lorimor, & van Hell [15] observed positivity at the 500-700ms time window when comparing English passive structures to active structures; this positivity was strongest at the left anterior electrodes. In a study of Japanese, Wolff, Schlesewsky, Hirotani, and Bornkessel-Schlesewsky [16] included sentences that were similar to those in German, in which the object-first structures were compared to subject-first structures via the use of a suffix in the nominative or accusative case on the NP. ERPs for object-initial compared to subject-initial structures after NP1 included an early (120-240ms) negativity, which was referred to as 'scrambling negativity', a broadly distributed positive shift at the NP1 of Object-Subject structures (400-650ms), an N400 at the NP2 for Subject-Object structures, and a late parietal negativity (650-1050ms) at the verb. Aside from the object scrambling, the positive shift at the NP1 in object-initial sentences was interpreted as the resolution of dependency introduced by an accusative-first argument. Another study on Japanese [17] investigated how context usage influences processing of object-first structures. They observed that object-first structures elicited a sustained LAN at NP1 and P600 at NP2 when the NP1 was new and not provided in the context.

A similar late parietal negativity at the verb was reported in Japanese in another study that used scrambled sentences [18]. A study on simple declarative sentences in Basque [19], which marks NPs with the ergative and absolutive cases, showed a similar negativity post-onset of the NP1 of object-first sentences. Like in Japanese, the NP2 follows the NP1 in Basque, with the verb in the final position. In the NP2 position, a left negativity (400-550ms) for object-first structures was found. In the P600 time window (700-900ms) in the verb position, a parietal positivity was observed. The negativity at the NP1 for object-first structures, although observed in a different time window, is suggested to be related to the scrambling negativity found in both German and Japanese. The effect at the NP2 is interpreted as a LAN that expresses working memory usage for displaced elements, or, alternatively, Erdocia et al. [19] suggested that

subjects and objects are processed differently regardless of their position. They hypothesized that the P600 observed at the verb position for object-first structures relates to an increase of processing costs when elements are displaced from their canonical positions. These studies attributed the late negativity to general increased processing of scrambled sentences.

For case-marking languages, there appears to be a pattern of a negative shift in NP1, which is the critical region of the sentence for thematic role assignment. However, other parts of the sentence indicate a different pattern: The P600 found in both Basque [19] and English [15] verbs is somewhat contradictory, as the case information denoting the thematic role assignment was provided earlier at the NP1 in Basque; as such, this P600 cannot be attributed to thematic role processing, and the authors attributed it to 'higher syntactic complexity'. By contrast, a study of Japanese [16] found a late negativity on the verb, which was an effect in a similar time window but with opposite polarity. Due to the inconsistencies in these findings (which may be in part attributed to the experimental design and the languages examined in these studies), it is particularly difficult to draw a conclusion on what constitutes a neural correlate of thematic role processing. To summarize the findings of comparable studies, Table 1 outlines the results of the discussed literature with results from [10] presented separately as unlike the other studies, they discuss relative clauses with different regions measured.

## Some relevant properties of Standard Indonesian

Indonesian is a zero-marking language [20] without case or gender markings. Transitive verbs are usually only inflected for voice (active or passive), and there is no verb inflection for tense, aspect, or agreement. Indonesian has SVO word order [21]; however, the ordering of constituents can be flexible, and it is possible (although infrequent) for verbs to take the initial position. Chung [22] suggested that Indonesian belongs to a branch of the Austronesian language family that was originally verb-initial, as the passivized transitive, active-transitive, as well as intransitive verbs can take the 1st position.

The usual transitive passive (5b) has the patient in the initial position. Examples of typical simple active and simple passive sentences are as follows:

5a) Simple active (agent-patient / SVO)

Perempuan   itu   **men**dorong   laki-laki   itu

girl   the   **ACT**-push   boy   the

'the girl is pushing the boy'

5b) Simple passive (patient-agent / OVS)

Laki-laki   itu   **di**dorong   (oleh)   perempuan   itu

boy   the   **PAS**-push   (by)   girl   the

'the boy is pushed by the girl'

The canonical sentence (5a) contains a verb with an active-transitive voice marking (*men-*). Likewise, the passive is expressed by the prefix (*di-*) on the verb in (5b), where 'the boy' is the patient of the action. Similar to English, the by-phrase is optional in the passive. In addition, the preposition (*oleh*) 'by' may be omitted when the agent is immediately adjacent to the verb (Cole & Hermon, 2008). As the current study observes the ERP distinctions between the simple active and the simple passive, it is also worth noting that the typical passive in Indonesian, unlike in most Indo-European languages, is highly frequent. It is acquired at a very early stage (around 2 years old; [23]) compared to English (4–5 years old), which can be attributed to its

**Table 1. Summary of previous ERP studies comparing word orders.**

| Language | Conditions | NP1 | NP2 | V | Note |
|---|---|---|---|---|---|
| German* [12] | SVO-OVS | LAN (400-600ms, 600-800ms) | - Negativity (400-1000ms) | Not discussed | Nom/Acc case was provided by articles preceding NPs. |
| | Ambiguity (fem. NP1 vs masc. NP1) | | - P600 (600-800ms, 800-1000ms) for amb. fem. NP1 | | |
| Japanese* [16] | SOV-OSV | - Scrambling negativity (120-240ms) | N400 (300-500ms) | Late negativity (650-1000ms) | Nom/Acc case was provided by markers following the NPs. |
| | | - positivity (400-650ms) | | | |
| Basque* [19] | SOV-OSV | Negativity (300-500ms) | Negativity (400-550ms) | P600 (700-900ms) | Erg/Abs case was provided by markers following the NPs. |
| English* [15] | Act-Pas | Not discussed | Not discussed | P600 (500-700ms) | Frontal distribution of P600- different from the typical distribution in garden-path sentences. |
| | Conditions | NP1 | RC onset | RC offset | |
| English [10] | S.RC–O.RC | Negativity (400-800ms) for reversible | Not found | positivity (-300-100ms) | Only found reversibility effects, no word order effect. |
| | Reversibility | (i.e. ani. NP1 vs inani. NP1) | | for rev. conditions | |

*all components are evoked to compare object-first to subject-first structures

high input frequency of 28–35% in Indonesian, compared to 4–5% in English. This difference is also reflected in the written form: Only 9% of English verbs display passive morphology [24] compared to 30–40% of Indonesian verbs [25] having the passive marker '*di-*'.

## The present study

The current study will investigate the processing of non-anomalous, simple sentences with differing word orders. The focus will be on the critical point in time, which is mainly the verb, but other regions of the sentence (such as NP2) will be investigated to check for spill-over effects.

The two conditions to be included are the active and the passive. The materials will be typical, plausible, reversible (both noun phrases are animate), violation-free sentences. There are some aspects of the Indonesian language that make the topic of the present study worth pursuing. First, the thematic roles of the NPs are coded by the passivization prefix on the verb rather than by case marking on the NPs. Second, unlike the studies of German [12] and Basque [19], no ambiguity manipulations are involved. Third, the structures to be tested in the current study are both relatively frequent compared to the passive structures in previously studied languages and are considered to be typical. The object-first conditions in the previous studies are infrequent in the respective languages compared to their subject-first counterparts; for example, the object-embedded relative clause in English [26] and the object-subject structure in German [27] are both highly infrequent structures. While arguments against exclusively syntactic frequency-based accounts of sentence processing have been established for German [28], there is behavioral evidence of the influence of sentence-level frequency and its interaction with other syntactic contrasts, such as the lexical bias of verbs [29–31].

The current experiment will contribute by providing electrophysiological data from Standard Indonesian, a zero-marking language that lacks many morphosyntactic features of Indo-European languages, paired with a relatively rigid word order. To the best of our knowledge, no ERP study has investigated Indonesian, where the 'non-canonical' sentence structure is relatively frequent. This is important because all the previous studies have compared one common sentence structure to an extremely infrequent one, and there is a wealth of evidence

suggesting that syntactic frequency plays a role in sentence processing at the behavioral level [2, 32, 33].

The outcome of the proposed experiment will reveal the neural correlates of thematic role assignment, which is a common process in sentence processing. The proposed study will attempt to determine *whether there are neural correlates at all* for thematic role processing of non-anomalous sentences and will reveal potential underlying cognitive processes required to parse sentences.

## Method

### Participants

50 right-handed, healthy, native speakers of Indonesian will participate, and be tested at the Hong Kong Polytechnic University. All the participants will be tested using an Indonesian translation of the short form of the Edinburgh Handedness Inventory [34] to ensure that they are classified as right-handed. Participants will provide informed consent and will be financially compensated. As a power analysis is not possible due to the lack of previously studies on non-anomalous sentence processing, we will use a heuristic estimate of the sample size needed by doubling the largest sample size of previous relevant studies we found on non-anomalous sentence processing: Jackson, Lorimor, & van Hell [15] had 25 participants, and therefore the proposed study will recruit 50 participants. We will recruit participants from both within the university (students and/or staff) as well as outside of it. The participants will have to complete a language background questionnaire to ensure that they have first-language or equivalent proficiency in Indonesian. This study was approved by the Hong Kong Polytechnic University's Institutional Review Board (ref no. HSEARS20211223003). Written consent will be obtained from participants.

### Materials

Participants will read stimuli comprising 100 semantically reversible active and passive sentences (see S1 Appendix in S1 File). Examples of which are shown in Table 2.

The sentences involve an animate subject and object and are constructed while avoiding a plausibility bias. Word order is manipulated by using two structures: simple active and simple passive sentences. Based on previous studies on processing well-formed sentences, the contrast of interest is passive-active, and whether the effects of the word order persist. We will strive for active-passive pairing plausibility with the use of NP1 and NP2 that are plausibly reversible (i.e. each sentence should not be strongly biased to one interpretation due to the nouns used). Additionally, there will be 200 filler trials consisting of questions (e.g. What did the adventurer notice yesterday?) and cleft sentences (e.g. it is the boy who the girl is calling) to prevent the habituation of the participants. The task for both the experimental and filler sentences will be discussed in the Procedure. The stimuli will be tested for acceptability and prototypicality using an online survey method utilizing a Likert scale of 1 to 7 with a similar setup used by Jackson, Lorimor, & van Hell [15]. Assuming the materials are rated acceptable and plausible by raters, the materials will not be further edited.

The materials will be presented visually and word-by-word for 500ms with a 100ms blank screen between each word. Each trial begins with a blank screen of 750ms followed by the word '*siap*?' (ready?), which stays on the screen until participants presses any keys. There will be two lists of sentences each pseudorandomized into 10 sets (each set containing 10 experimental sentences and 20 fillers). Digital triggers will be manually inserted at three time points in every sentence: at the onset of the verb (i.e., the onset of the prefix), and at the onset of NP2. Additionally, NP1 data points will be extracted from -1400 to -200 before the verb onset to be

**Table 2. Stimuli examples of each condition.**

| Condition | NP1 | Art | VP | Adjunct | NP2 | PP/RC |
|---|---|---|---|---|---|---|
| Active | Polisi | itu | menembak | langsung | seorang perampok | di malam hari. |
| | Police | that/the | **ACT**shoot | immediately | (a) robber | at night. |
| | | (the/a) police immediately shoots (the/a) robber at night. | | | | |
| Passive | Polisi | itu | ditembak | langsung | oleh perampok | di malam hari. |
| | Police | that/the | **PAS**shoot | immediately | by robber | at night |
| | | (the/a) police is immediately shot by (the/a) robber at night | | | | |

used as a sanity check to see if there are baseline differences between the conditions. It is important to note that the triggers will be located in the middle of the sentence.

## Frequency

The main aim of this study is to essentially 'control' for the possible effect of frequency. It might not be feasible to manipulate the relative syntactic frequency of a structure in a language. However, at the lexical level, verb bias (or the frequency with which verbs appear in different structures) influences both production [35] and comprehension [36]. While most of the previous studies did not address this variable in their materials, one study by Jackson et al. [15] attempted this by using the progressive participle '-ing' in English (which is less frequent than the simple active verb form and is somewhat comparable to the passive) in combination with the past tense instead of simple present active structures. This has two potential issues; the first is we interpret thematic role processing as the assignment of agent/patient role to the first/second noun phrase in the study, and the active voice has several verb forms such as the simple form in which the first noun phrase is the agent, and when considered as a collective as they should be, these occur more frequently than the passive form. The second issue is whether the additional progressive aspect information deviating from the canonical default of the simple present active compared to passive will elicit a different neural signal is unknown. In our study, we will attempt to control verb bias by incorporating verb pairs that fulfil either of the two criteria:

1. A verb with higher token frequency in its passive form compared to its active form.

2. A verb with the same frequency class* for both the passive and the active form.

*Frequency class is a number assigned to a group of words whereby this number does not often change in different corpora. The calculation is as follows: The frequency of the most frequent word in the corpus is divided by the frequency of the specific word, and log base 2 of the result is rounded up to the closest whole number.

We used the *Indonesian mixed corpus*, which is the largest Indonesian online corpus in the Leipzig Corpora Collection [37]. The distribution and descriptive information of the frequency can be seen in Fig 1 and Table 3, respectively (for full frequency information about the verbs, see S2 Appendix in S1 File). In Fig 2, log-transformed frequency values of active verbs are deducted by log-transformed frequency values of passive verbs. The general pattern is not only are they relatively comparable, but if anything, the passive verbs are slightly more frequent for some pairs. Extremely rare words (with a frequency class of 20 or higher) will not be used in this study, as the highest frequency class is 17 and the lowest is 7.

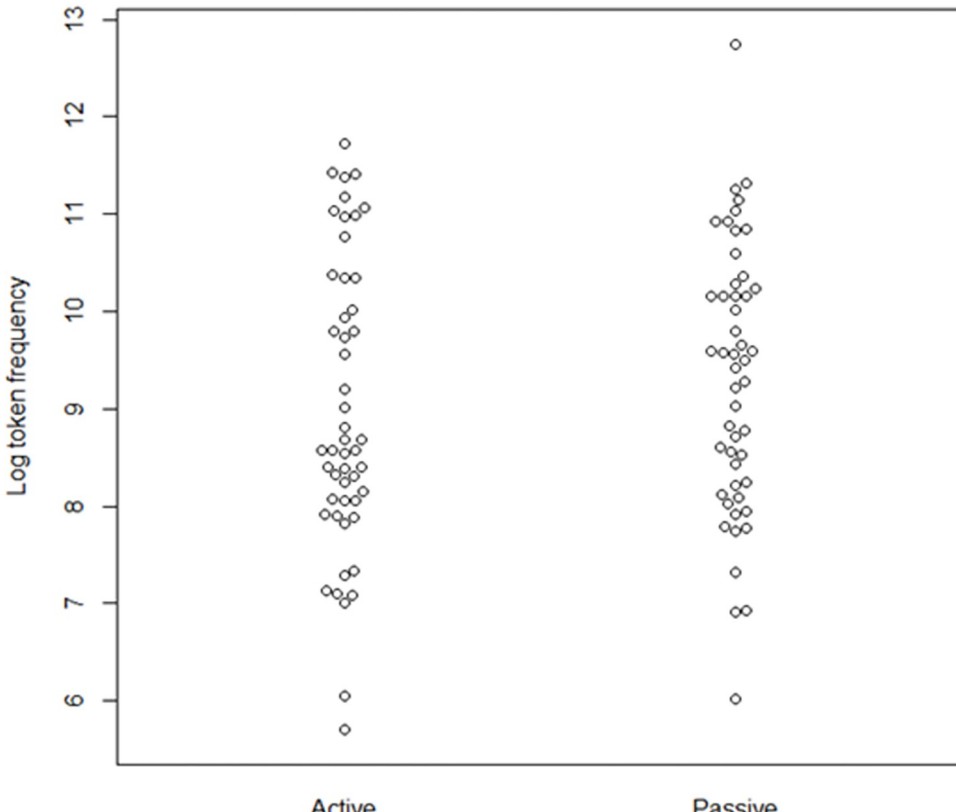

**Fig 1. Beeswarm plot of log-transformed passive and active verb frequencies.**

## Procedure

Participants will read the participant information sheet, fill out a questionnaire about their demographic details, and sign the informed consent form.

Participants will be seated in front of a presentation monitor on which the sentences will be presented using E-Prime software, and the experiment will start with written instructions that will be explained orally by the experimenter. During the whole experiment, a fixation cross will be shown between trials and sets. Prior to the experiment, participants will be instructed to minimize their head movements during the trials and not to close their eyes while performing the experiment, although they will not be asked to refrain from blinking.

After every 3–8 trials (randomized), participants will be given a comprehension prompt. For the experimental items and filler cleft sentences, the comprehension prompt will be a yes/no question to probe the thematic role assignment (e.g., Did the police shoot the robber?/Did

**Table 3. Descriptives of verb frequency information.**

|  | Active | Passive |
|---|---|---|
| Mean | 21513.1 | 26415.88 |
| SD | 30110.56 | 50517.23 |
| *Range* | 124326 | 342184 |
| *Min* | 301 | 411 |
| *Max* | 124627 | 342595 |

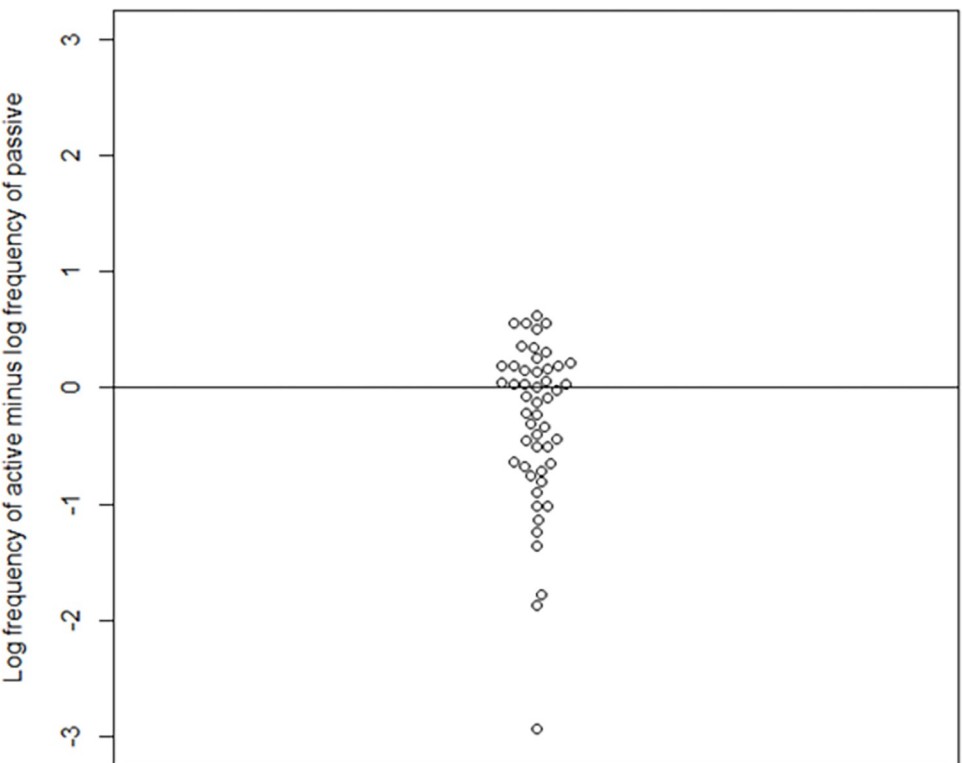

**Fig 2. Beeswarm plot of log-transformed difference between active and passive (active–passive).**

the robber shoot the police?). For the filler questions, the comprehension prompt will be a sentence that appropriately responds to the question (e.g., the adventurer noticed an officer patrolling) or if the noun phrase in the question is presented as the theme (e.g., the officer called the adventurer), and participants will have to judge whether the sentence is an appropriate response to the question (e.g., they should respond "yes" if they see "the adventurer noticed an officer patrolling" after this filler). These prompts are designed to ensure that the participants remain focused and parse the trials displayed, as well as to provide a measure of comprehension performance.

To avoid fatigue, there will be pauses after every block consisting of 30 trials (10 blocks in total), where the participants can take a break and resume the experiment whenever they are ready. The total test session, including cap and electrode preparation, will take approximately one and a half hours per participant.

## EEG recording and preprocessing

EEGs will be recorded using 64 Ag-Acl electrodes that will be attached to the participant's scalp via an elastic cap with a 10–20 system. Conductive gel will be used. The cap has two dedicated electrodes for the left and right mastoids. To monitor horizontal and vertical eye movements, two electrodes are fixed in the outer canthi of each eye, and one more will be placed below the left eye (the VEOG above the left eye is integrated in the cap). Electrode impedances will be kept below 5k Ω. The EEG will be amplified and digitized with a sampling rate of 1000 Hz with an analog bandpass filter of 0.03-100Hz. The amplifier that will be used is the SynAmps 2 (NeuroScan, Charlotte, NC, United States), and the cap will be the 64-channel Quik-Cap Neo Net (NeuroScan, Charlotte, NC, United States). Stimtracker (Cedrus) will

provide an interface between the experiment presentation software and EEG acquisition. Continuous EEG data will be acquired using Curry 7 acquisition software (Compumedics NeuroScan) whereby the files will be exported to the.cnt format and analysed using EEGLAB [38] for the preprocessing, and FieldTrip [39] for the statistical analysis.

The EEG data will be re-referenced to the two mastoid electrodes. The ERPs will be calculated per participant, per electrode, and per condition in intervals of 200ms before onset to 1000ms after onset for each time-locked trigger. These epochs will then be demeaned per channel in each epoch (the mean of the data from the entire epoch will be subtracted from each data point, as this may result in better ICA decompositions than baseline-correcting based on pre-stimulus interval; [40]. The epochs will then be subjected to an independent component analysis using the runica() command in EEGLAB [41]; this will divide the data into many independent components corresponding to the number of channels, excluding mastoid electrodes, EOGs, and bad channels that are previously marked. These components will be inspected visually to identify blinks and saccades, and components that are identified as blinks or saccades will be removed (a maximum of four components per participant can be removed, and if the number of components removed exceeds this, the participant will be excluded from the analysis). After the removal of components corresponding to blinks and saccades, baseline correction will be applied to the data with a 200 ms pre-stimulus onset baseline. Next, the epochs will be run through a moving window peak-to-peak threshold function for artefact detection; epochs with artifacts will be marked for removal based on this criterion. Lastly, the data will be filtered using a 0.1–40.0 Hz bandpass filter via the pop_eegfiltnew() default EEGLAB function.

There is an alternative method of *not* using baseline corrections, as suggested by Wolff et al. [16] and Friederici, Wang, Herrmann, Maess, & Oertel [42] who conducted sentence processing experiments. The reasoning behind this approach is that in the mid-sentence time windows, the waves of each trial may not be identical prior to the onset of the critical word, therefore potentially distorting the baseline. Wolff et al. [16] instead used narrower bandpass filters (0.3–20.0Hz) to exclude slow drifts while still including language-related ERPs. However, Steinhauer [43] criticized the use of a higher filter instead of baseline-correction because first of all, the modified filter does not distinguish between artifacts (slow drifts) and real slow waves related to language processing; moreover, the filter converts sustained effects into apparent local effects such as ELANs, and finally, the increased filtering does not directly address the problem of a distorted baseline resulting from differences before the onset of the critical region. We therefore will adopt a 200ms baseline in this study.

## Data exclusion criteria

Participants may be excluded based on our predetermined data exclusion criteria, which include the following:

- Minimum number of trials after artifact rejection: We determine a preset number of 25 trials per condition as a minimum. Any participant having fewer than 25 trials in any condition after artifact rejection will be excluded from the analysis.

- Missing information/data: If a participant refuses to complete all or part of the questionnaire about her/his demographic information or the handedness inventory, or if a technical issue leads to missing data/a subset of missing data for a participant, the individual will be excluded from the analysis.

- Bad channels (1): The threshold for data exclusion is at or over 15% of the electrodes (9 or more) being unusable due to excessive artifacts or environmental noise. This is in the event

that these channels cannot be interpolated due to positioning (for example, multiple bad channels being adjacent to each other and therefore not having enough neighbouring electrodes for interpolation).

- Bad channels (2): A second criterion is if a number of bad channels cluster or are adjacent to one another: If there are 6 bad channels in one cluster/adjacent to one another, the participant will be excluded.

- Signal-to-noise ratio: As proposed by Parks, Gannon, Long, & Young [44], bootstrap resampling with a value of 9999 bootstraps of ERP waveforms will be used to calculate a signal-to-noise ratio confidence interval for each individual subject (a minimum of 25 trials). The lower bound of this calculation provides an objective measure of signal quality, and ensures that the components we are analyzing exceed a predetermined signal-to-noise ratio. Participants whose signal-to-noise ratio lower bound is below 3.0 dB will be excluded from the analysis. This specific value is within the accuracy peak of subject classification, and experienced ERP investigators have generally considered waveforms below 3.0 dB to be noisy when they are visually inspected.

## Statistical analysis plan

The statistical analysis will be conducted using cluster-based permutation tests [45] over all the scalp electrodes and the entire post-stimulus epoch. The advantage of this approach is that it allows testing for effects anywhere on the scalp and any time in the epoch, while still controlling the familywise false positive rate, and without the experimenter needing to choose regions and time windows for analysis. The test works by comparing the active and passive ERPs at each channel and each sample and identifying clusters of spatiotemporally adjacent data points where the difference between the two conditions exceeds some threshold; in our analysis, that threshold will be two-tailed $p < .1$ in a t-test. In other words, a t-test comparing active and passive will be performed at every sample in every channel, if a series of several time points in a row on the same channel and/or several adjacent channels at the same time all exceed this threshold, they are treated as a "cluster". Next, each cluster is assigned a test statistic (in our case, the test statistic for a cluster is derived by summing the $t$-values of all the samples in the cluster), and the largest cluster-level test statistic in the epoch is taken as the observed test statistic for the data. Next, the data are randomly permuted (i.e., within each subject, the condition labels "active" and "passive" may be randomly switched) several thousand times, and with each random permutation the abovementioned procedure of identifying clusters and calculating a test statistic is repeated. This yields a permutation distribution of several thousand test statistics, against which the original observed test statistic is compared. The proportion of permutation test statistics that are larger than the original observed test statistic is the $p$-value for the test; if there is a significant difference between the active and passive ERPs then this value will be small. See [45] for a more detailed explanation of how this test works.

We will run the test using a cluster threshold of $p < .1$, as this makes the test more sensitive to weak, sustained effects similar to what was found in previous studies that have investigated thematic role processing (see [45], for discussion of how the cluster threshold influences the sensitivity of the test to different types of effects) and at least two spatial neighbouring electrodes that also meet the threshold (we will use the *minnbchan = 2* function in the Fieldtrip implementation of the cluster-based test). The permutation test will use 5000 iterations.

## Predictions

There are several possible patterns of the results. These are described below.

1. An ERP contrast can be expected between canonical and non-canonical sentences on the critical region (verb). We can expect a posterior positivity (P600) at the disambiguation point to indicate that a revision of the thematic information is needed: NP1 should be reassigned from the default agent role to become the patient of the action.

2. An ERP contrast can be expected between canonical and non-canonical sentences in the NP2 time windows. Additionally, based on earlier studies we expect a 'generalized' increase in processing costs in comprehending the non-canonical structures attributed to violation of transitivity expectation [16] as well as retrieval of verbal material in a non-canonical position and uptick in working memory demands [19]. For example, the verbs in Basque [19] and Japanese [16] provide no additional thematic information, but within that section, ERP differences for object-first compared to subject-first structures were observed.

3. A lack of ERP correlates in both the verb and NP2, especially between the passive and active conditions, suggests that previous findings regarding the neural correlates of thematic role assignment were confounded by syntactic frequency because the studies compared one highly frequent structure (e.g. active) to a highly infrequent sentence structure (e.g. passive).

In general, for the time windows and specific components, due to the nature of cluster-based permutation where adjacent sites and time points are correlated, we will be looking at the whole epoch (here we theorize that a 'real' effect should persist through multiple adjacent electrodes and a chunk of tens to hundreds of milliseconds/samples–a more detailed description of the procedure is provided by [45]). However, were there to be an effect in either the verb or NP2 (as per the predictions above), we expect it to be observed in the time windows and distributions corresponding to the LAN, N400, and/or P600 –all of which have been observed in similar studies and above 300ms. Specifically, for the verb, we would expect the passive to be more positive than the active for the 500-700ms time window, as this was what the study closest [15] to this one reported in their critical region. For NP2, we expect some form of negativity to occur for the passive compared to the active between the 300-600ms time window–this was observed in studies which reported ERPs post-disambiguation.

## Results

### Timeline

Funding and research staff have already been secured for this project. With the assumption that this *Registered Report Protocol* is accepted, we hope to carry out data collection during fall 2022-or as soon as we receive an in-principle acceptance notice, complete the data analysis within winter 2022 (as most of the data processing steps described in the manuscript are automated, this step can be completed very quickly), and we will submit the *Registered Report* by spring 2023. One potential issue here is that our institution is in a location where Indonesian is not the first language for most of the population. However, we believe we will still be able to recruit a sufficient number of participants from the immigrant and student population.

## Supporting information

**S1 File.**
(DOCX)

## Author Contributions

**Conceptualization:** Bernard A. J. Jap, Yu-Yin Hsu.

**Data curation:** Bernard A. J. Jap.

**Formal analysis:** Bernard A. J. Jap, Stephen Politzer-Ahles.

**Funding acquisition:** Bernard A. J. Jap, Yu-Yin Hsu, Stephen Politzer-Ahles.

**Investigation:** Bernard A. J. Jap, Yu-Yin Hsu, Stephen Politzer-Ahles.

**Methodology:** Bernard A. J. Jap, Yu-Yin Hsu, Stephen Politzer-Ahles.

**Project administration:** Bernard A. J. Jap, Stephen Politzer-Ahles.

**Resources:** Yu-Yin Hsu, Stephen Politzer-Ahles.

**Software:** Stephen Politzer-Ahles.

**Supervision:** Yu-Yin Hsu, Stephen Politzer-Ahles.

**Validation:** Bernard A. J. Jap.

**Visualization:** Bernard A. J. Jap.

**Writing – original draft:** Bernard A. J. Jap, Yu-Yin Hsu, Stephen Politzer-Ahles.

**Writing – review & editing:** Bernard A. J. Jap, Yu-Yin Hsu, Stephen Politzer-Ahles.

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
