## [Decision Letter · Decision Letter 0]

25 Apr 2022

PONE-D-22-01301Neural Correlates of Thematic Role Assignment for Passives in Standard IndonesianPLOS ONE

Dear Dr. Jap,

Thank you for submitting your manuscript to PLOS ONE. After careful consideration, we feel that it has merit but does not fully meet PLOS ONE’s publication criteria as it currently stands. Therefore, we invite you to submit a revised version of the manuscript that addresses the points raised during the review process.

The study has been evaluated by two independent Reviewers that found your study novel and worth to be pursued. They also raised a number of issues that I would like to be addressed before acceptance. please consider them carefully providing a point-by-point response.

We look forward to receiving your revised manuscript.

Kind regards,

Nicola Molinaro, Ph.D.

Academic Editor

PLOS ONE

Journal Requirements: 

Reviewers' comments:

Reviewer's Responses to Questions

**Comments to the Author**

1. Does the manuscript provide a valid rationale for the proposed study, with clearly identified and justified research questions?

Reviewer #1: Yes

Reviewer #2: Yes

2. Is the protocol technically sound and planned in a manner that will lead to a meaningful outcome and allow testing the stated hypotheses?

Reviewer #1: Yes

Reviewer #2: Partly

3. Is the methodology feasible and described in sufficient detail to allow the work to be replicable?

Reviewer #1: Yes

Reviewer #2: Yes

4. Have the authors described where all data underlying the findings will be made available when the study is complete?

Reviewer #1: No

Reviewer #2: No

5. Is the manuscript presented in an intelligible fashion and written in standard English?

Reviewer #1: Yes

Reviewer #2: Yes

6. Review Comments to the Author

You may also provide optional suggestions and comments to authors that they might find helpful in planning their study.

Reviewer #1: SUMMARY

The authors report the design and motivation of an ERP study in standard Indonesian compare active and passive structures. The authors explain the rationale for the study, namely that previous ERP studies of canonical vs. noncanonical structures are often confounding structure with frequency, whereas in Indonesian the passive is much more frequent on average (and, for some verbs, apparently the preferred structure). This allows the authors to ascertain whether the standard ERP effects of grammatical, noncanonical structures relative to grammatical, canonical ones is due to the underlying structural alternation or frequency. The authors detail the design, involving 100 active/passive sentences along with 100 filler structures, including detailed EEG data collection procedure and analysis pipeline.

EVALUATION

The study is well-motivated and the experimental design is rigorous. The manuscript is clearly written. I have no major concerns, and look forward to seeing the results of the study. As a side comment, I appreciate that the inclusion of such an under-studied language is nicely scientifically motivated, given the gaps in the existing literature. I do have some minor comments below.

MINOR

I did not notice where the data will be made available upon study completion.

The discussed on animacy on page 8, lines 174-176 seemed unnaturally truncated, with the sentence about Meltzer and Braun (2013) feeling quite out of context and unfinished.

Table 1 was a bit unclear; the study by Meltzer and Braun (2013) is set off with different headers, but not much explanation is provided to understand what this means. In general, the whole table needs to be fleshed out more to indicate what to expect in each column of the listed studies.

It might be worth considering additional fillers beyond the question structure to draw attention away from the critical active/passive structures in declarative sentences.

It might be worth inserting triggers for every word position in order to allow other possible analyses, e.g. a “sanity check” at NP1 in order to indicate that there are no baseline differences between conditions, etc. Why not just use triggers for every position (just in case)?

page 5, examples 2 and 3 are translated the same way, but as I understand it, the roles should be reversed between these two sentences given the nominative case on ‘guitar player’ in (2) and accusative case on ‘guitar player’ in 3.

page 16, line 367 – typo, I believe “shown on the in between trials” should probably be “shown on the screen in between trials”

Reviewer #2: The manuscript proposes a study with the goal to investigate the involved processes as indexed by ERP components when reading well-formed sentences in active and passive structures. From the provided document, their main goals are to 1) identify ERP components indexing thematic role assignment (or find that such a component does not exist) and 2) disentangle the various components for various languages reported in the literature. Novel to prior research, their study investigates Indonesian as a language for which the passive structure is much more commonly used as in the previously investigated languages, thus, eliminating frequency effects that might have tainted previous research.

This approach of disentangling the effects of thematic role assignment from the potentially confounding frequency effect appears reasonable and highly interesting.

However, I would like to share some concerns and notes that arose when reading the manuscript:

Major comments:

1) According to the guidelines, the authors are supposed to describe where the data will be available. While the authors state that they are planning to make the data available, they do not mention where.

2) The authors mention that the stimuli will be pretested for acceptability. In order to avoid a potential confound, I would suggest also pretesting for prototypicality. In their example in Table 2, the police is the patient of the shooting while the robber is the agent. This seems potentially unproblematic, but judging from the material list, there might be cases where there is a preferred role assignment based on the involved entities. Depending on whether this preference is met or not, potential effects could be evoked.

3) While the authors convincingly explain their planned method of data processing starting in line 425, I see potential issues with the baseline correction approach, especially in regards to the NP2. As the NP2 is immediately following the Verb, the baseline correction will be conducted on potential effects in the verb region. This is especially an issue because a P600 effect is even predicted for the verb region. This issue is also directly related with another concern: The presentation rate (500 ms + 100 ms) puts the onset of the NP2 directly into the P600 time-window of the verb. The epoch length for the planned ERP analysis with 1000 ms after stimulus onset thus also contains the onset of the NP2. This might pose a problem for both time-windows: the P600 could be shifted by early effects evoked by the NP2 while early effects on the NP2 could suffer from the underlying wave form of the P600. In that regard, I was wondering if Indonesian allows to fill the position between the verb and NP2 with e.g., temporal adverbs or similar that could be kept constant across conditions in order to create a longer distance between the verb and NP2 and, thus, minimize the overlap of effects from different regions and the baseline correction issue.

4) Related to the previous point, I was wondering about the necessity to keep the semantic content of the passive and active structures identical. Currently, the swap from active and passive leads to a swap of the NP2 as well, because the “robber” (from the example) maintains the role of the shooter. This leads to a target manipulation which does not seem particularly necessary in this case, given that both entities similarly reasonably could take the active and passive role in the sentence. Judging from the material section, it appears as though the authors paid attention to follow this premise. This also again ties in with the earlier mentioned pretest. I believe that, in general, avoiding a target manipulation, when possible, should be preferred. While I understand concerns regarding the change of the semantic content of the sentence, it appears to me that this would be possible here in favor of “cleaner” comparisons.

5) The Predictions section should be fleshed out more. While the authors specifically state predictions for the P600 component in the verb region, for the NP2 region they only mention a “generalized” increase of processing cost. Also, it is not clear to me whether prediction 3 (l. 505) is stated in regard to the NP2, the verb or both.

Given their literature review, it would be nice to see the predictions again being put into that context. Which time-windows are they considering? In favor of which interpretation/research would a component (or the lack of a component) be?

Lastly, a comparably minor comment in relation to the predictions: The authors acknowledge the difference between certain languages, as for example that Basque and Japanese do not provide thematic information on the verb, while Indonesian entirely provides this information on the verb. As the region which allows for thematic assignment thereby shifts and might be entangled with different processes that are inherent to the respective region, a direct comparison of effects (or the lack of effects) might be a bit problematic. I agree that their planned study could highly contribute to the field and would yield valuable new insights, but a direct comparison should still be done with caution.

Minor Comments:

1) As the authors discussed research on Japanese, I was thinking of a more recent study by Yano and Koizumi (Yano, M., & Koizumi, M. (2018). Processing of non-canonical word orders in (in) felicitous contexts: Evidence from event-related brain potentials. Language, Cognition and Neuroscience, 33(10), 1340-1354.). While they investigate the effect of givenness in combination with word order, their findings might potentially enrich the discussion of the current study and provides a more recent reference to the study of word order in Japanese. Especially, as a focus of the literature review was put on studies on well-formed sentences.

2) I am not very familiar with the means of analysis planned by the authors. From my understanding, the cluster-level test statistic is not the standard in the literature. As such, I think it would be helpful if this section could be expanded a bit more.

3) The sentence staring at line 348 appears to be erroneous.

4) The sentence in line 367 is missing a word (“a fixation cross will be shown on the in between trials and sets.”)

7. PLOS authors have the option to publish the peer review history of their article (what does this mean?). If published, this will include your full peer review and any attached files.

Reviewer #1: **Yes: **William Matchin

Reviewer #2: No

---

## [Author Response · Author response to Decision Letter 0]

8 Jun 2022

We would like to thank the editor and reviewers for their helpful feedback; this has helped us make the issues at stake in the paper clearer. All changes in the manuscript are highlighted yellow. Our detailed responses to the reviewers are below.

Editor Comments

 We have revised the style to match the formatting samples.

We have added an additional description of the data availability for our study after the timeline (p26). 

Reviewer 1

The authors report the design and motivation of an ERP study in standard Indonesian compare active and passive structures. The authors explain the rationale for the study, namely that previous ERP studies of canonical vs. noncanonical structures are often confounding structure with frequency, whereas in Indonesian the passive is much more frequent on average (and, for some verbs, apparently the preferred structure). This allows the authors to ascertain whether the standard ERP effects of grammatical, noncanonical structures relative to grammatical, canonical ones is due to the underlying structural alternation or frequency. The authors detail the design, involving 100 active/passive sentences along with 100 filler structures, including detailed EEG data collection procedure and analysis pipeline.

EVALUATION

The study is well-motivated and the experimental design is rigorous. The manuscript is clearly written. I have no major concerns, and look forward to seeing the results of the study. As a side comment, I appreciate that the inclusion of such an under-studied language is nicely scientifically motivated, given the gaps in the existing literature. I do have some minor comments below.

MINOR

I did not notice where the data will be made available upon study completion.

Thanks for pointing this out, we have added an additional subsection after the timeline (p24).

The discussed on animacy on page 8, lines 174-176 seemed unnaturally truncated, with the sentence about Meltzer and Braun (2013) feeling quite out of context and unfinished.

This part has been removed; the study has been discussed in an earlier section (p6).

Table 1 was a bit unclear; the study by Meltzer and Braun (2013) is set off with different headers, but not much explanation is provided to understand what this means. In general, the whole table needs to be fleshed out more to indicate what to expect in each column of the listed studies.

Thank you for the feedback. We added an explanation before Table 1 (p8) on its contents and why the Meltzer and Braun (2013) study has a separate header in the table.

It might be worth considering additional fillers beyond the question structure to draw attention away from the critical active/passive structures in declarative sentences.

We have added another set of fillers using cleft sentences bringing the total to 300 sentences. The comprehension question for this set will be similar to the experimental items (who did what to who). While this will increase experiment duration, we believe that the original experiment duration is already relatively short, and this addition will not affect the quality of the data from concerns such as the gel drying out or waning concentration of the participants (previous estimated duration: 28 minutes 45 seconds, current estimated duration: 43 minutes and 3 seconds). This information is added on page 14 & 19.

It might be worth inserting triggers for every word position in order to allow other possible analyses, e.g. a “sanity check” at NP1 in order to indicate that there are no baseline differences between conditions, etc. Why not just use triggers for every position (just in case)?

This is a fantastic suggestion, thank you very much. We will analyze the data time-locked to NP1 (this information is added on p14). 

page 5, examples 2 and 3 are translated the same way, but as I understand it, the roles should be reversed between these two sentences given the nominative case on ‘guitar player’ in (2) and accusative case on ‘guitar player’ in 3.

page 16, line 367 – typo, I believe “shown on the in between trials” should probably be “shown on the screen in between trials”

 Both errors have been amended.

Reviewer 2

The manuscript proposes a study with the goal to investigate the involved processes as indexed by ERP components when reading well-formed sentences in active and passive structures. From the provided document, their main goals are to 1) identify ERP components indexing thematic role assignment (or find that such a component does not exist) and 2) disentangle the various components for various languages reported in the literature. Novel to prior research, their study investigates Indonesian as a language for which the passive structure is much more commonly used as in the previously investigated languages, thus, eliminating frequency effects that might have tainted previous research.

This approach of disentangling the effects of thematic role assignment from the potentially confounding frequency effect appears reasonable and highly interesting.

However, I would like to share some concerns and notes that arose when reading the manuscript:

Major comments:

1) According to the guidelines, the authors are supposed to describe where the data will be available. While the authors state that they are planning to make the data available, they do not mention where.

We have added an additional subsection after the timeline referring to data availability (p26). 

2) The authors mention that the stimuli will be pretested for acceptability. In order to avoid a potential confound, I would suggest also pretesting for prototypicality. In their example in Table 2, the police is the patient of the shooting while the robber is the agent. This seems potentially unproblematic, but judging from the material list, there might be cases where there is a preferred role assignment based on the involved entities. Depending on whether this preference is met or not, potential effects could be evoked.

Thank you for the suggestion. We will test prototypicality along with acceptability (p14).

3) While the authors convincingly explain their planned method of data processing starting in line 425, I see potential issues with the baseline correction approach, especially in regards to the NP2. As the NP2 is immediately following the Verb, the baseline correction will be conducted on potential effects in the verb region. This is especially an issue because a P600 effect is even predicted for the verb region. This issue is also directly related with another concern: The presentation rate (500 ms + 100 ms) puts the onset of the NP2 directly into the P600 time-window of the verb. The epoch length for the planned ERP analysis with 1000 ms after stimulus onset thus also contains the onset of the NP2. This might pose a problem for both time-windows: the P600 could be shifted by early effects evoked by the NP2 while early effects on the NP2 could suffer from the underlying wave form of the P600. In that regard, I was wondering if Indonesian allows to fill the position between the verb and NP2 with e.g., temporal adverbs or similar that could be kept constant across conditions in order to create a longer distance between the verb and NP2 and, thus, minimize the overlap of effects from different regions and the baseline correction issue.

That is a very good point. We have edited the materials to incorporate an adjunct (using various adverbs such as yesterday, immediately, continuously that collocate with the verb) between the verb and NP2 to allow for a more generous baseline for NP2 and a bigger analysis time window for the verb. The changes are shown on p13 and on the Appendices p27-31.

The stimulus is now as follows:

Condition NP1 Art VP Adjunct NP2 PP/RC

Active 

 Polisi itu menembak langsung perampok di malam hari. 

 Police that/the ACTshoot immediately robber at night. 

 (the/a) police immediately shoots (the/a) robber at night.

Passive Polisi itu ditembak langsung perampok di malam hari. 

 Police that/the PASshoot immediately robber at night 

 (the/a) police is immediately shot by (the/a) robber at night

4) Related to the previous point, I was wondering about the necessity to keep the semantic content of the passive and active structures identical. Currently, the swap from active and passive leads to a swap of the NP2 as well, because the “robber” (from the example) maintains the role of the shooter. This leads to a target manipulation which does not seem particularly necessary in this case, given that both entities similarly reasonably could take the active and passive role in the sentence. Judging from the material section, it appears as though the authors paid attention to follow this premise. This also again ties in with the earlier mentioned pretest. I believe that, in general, avoiding a target manipulation, when possible, should be preferred. While I understand concerns regarding the change of the semantic content of the sentence, it appears to me that this would be possible here in favor of “cleaner” comparisons.

Thank you for the feedback. Indeed, if prototypicality is not an issue there would not be grounds to reverse the roles of the NP2 between the conditions. However, we aimed to make this as comparable as possible to previous studies, and one which we modeled this study upon is the study in English by Jackson, Lorimor, & van Hell (2020) – here they kept the semantic content identical. 

5) The Predictions section should be fleshed out more. While the authors specifically state predictions for the P600 component in the verb region, for the NP2 region they only mention a “generalized” increase of processing cost. Also, it is not clear to me whether prediction 3 (l. 505) is stated in regard to the NP2, the verb or both.

Given their literature review, it would be nice to see the predictions again being put into that context. Which time-windows are they considering? In favor of which interpretation/research would a component (or the lack of a component) be?

The prediction subsection has been expanded to improve clarity (p24-25). As you’ve mentioned in the following comment, it can be difficult to interpret a direct comparison between the findings of studies on case-marking languages and Indonesian (e.g. would the NP2 in Indonesian correspond with NP2 in Japanese/Basque, given the thematic information is provided in the preceding section of the sentence [V in Indonesian and NP1 in Japanese/Basque]?). However, we have added some additional information (p24) in the prediction to show what we can expect to find if the findings turn out to be comparable to the results of case-marking languages. As for the time windows and specific components, due to the nature of cluster-based permutation where adjacent sites and time points are correlated, we will be looking at the whole epoch (here we theorize that a ‘real’ effect should persist through multiple adjacent electrodes and a chunk of tens to hundreds of milliseconds/samples – a more detailed description of the procedure is provided by Maris and Oostenveld, 2007). However, were there to be an effect in either the verb or NP2, we expect it to be observed in the time windows and distributions corresponding to the LAN, N400, and/or P600 – all of which have been observed in similar studies and above 300ms. Specifically, for the verb, we would expect the passive to be more positive than the active for the 500-700ms time window, as this was what the study closest (Jackson et al., 2020) to this one reported in their critical region. For NP2, we expect some form of negativity to occur for the passive compared to the active between the 300-600ms time window – this was observed in studies which reported ERPs post-disambiguation.

Lastly, a comparably minor comment in relation to the predictions: The authors acknowledge the difference between certain languages, as for example that Basque and Japanese do not provide thematic information on the verb, while Indonesian entirely provides this information on the verb. As the region which allows for thematic assignment thereby shifts and might be entangled with different processes that are inherent to the respective region, a direct comparison of effects (or the lack of effects) might be a bit problematic. I agree that their planned study could highly contribute to the field and would yield valuable new insights, but a direct comparison should still be done with caution.

This is indeed challenging, and we plan to interpret the findings with care in regard to discussing it in the light of previous studies on languages with case marking.

Minor Comments:

1) As the authors discussed research on Japanese, I was thinking of a more recent study by Yano and Koizumi (Yano, M., & Koizumi, M. (2018). Processing of non-canonical word orders in (in) felicitous contexts: Evidence from event-related brain potentials. Language, Cognition and Neuroscience, 33(10), 1340-1354.). While they investigate the effect of givenness in combination with word order, their findings might potentially enrich the discussion of the current study and provides a more recent reference to the study of word order in Japanese. Especially, as a focus of the literature review was put on studies on well-formed sentences.

Thank you for the input. We have incorporated the findings into the manuscript (p7).

2) I am not very familiar with the means of analysis planned by the authors. From my understanding, the cluster-level test statistic is not the standard in the literature. As such, I think it would be helpful if this section could be expanded a bit more.

We have revised the section to clarify the analysis procedure further (p23-24)

3) The sentence staring at line 348 appears to be erroneous.

4) The sentence in line 367 is missing a word (“a fixation cross will be shown on the in between trials and sets.”)

Both errors have been amended. (p15;p18)

---

## [Decision Letter · Decision Letter 1]

20 Jun 2022

PONE-D-22-01301R1Neural Correlates of Thematic Role Assignment for Passives in Standard IndonesianPLOS ONE

Dear Dr. Jap,

Thank you for submitting your manuscript to PLOS ONE. After careful consideration, we feel that it is essentially ready to be accepted for publication -- there's just a clarification point raised by Reviewer 1 (and Reviewer 2 suggested that some of your points were better articulated in the letter than in the manuscript). Therefore, we invite you to submit a revised version of the manuscript that fixes this.

We look forward to receiving your revised manuscript.

Kind regards,

Daniel Mirman

Academic Editor

PLOS ONE

Journal Requirements:

Reviewers' comments:

Reviewer's Responses to Questions

**Comments to the Author**

1. Does the manuscript provide a valid rationale for the proposed study, with clearly identified and justified research questions?

Reviewer #1: Yes

Reviewer #2: Yes

2. Is the protocol technically sound and planned in a manner that will lead to a meaningful outcome and allow testing the stated hypotheses?

Reviewer #1: Yes

Reviewer #2: Yes

3. Is the methodology feasible and described in sufficient detail to allow the work to be replicable?

Reviewer #1: Yes

Reviewer #2: Yes

4. Have the authors described where all data underlying the findings will be made available when the study is complete?

Reviewer #1: Yes

Reviewer #2: Yes

5. Is the manuscript presented in an intelligible fashion and written in standard English?

Reviewer #1: Yes

Reviewer #2: Yes

6. Review Comments to the Author

You may also provide optional suggestions and comments to authors that they might find helpful in planning their study.

Reviewer #1: I appreciate the revisions and think it essentially looks good, but I still am confused about sentence examples (2) and (3) on page 5.

The provided English translations are identical for these two sentences ~"the guitar player discovered the singer", implying that the semantic roles are identical in the two sentences. But, as I understand it, the case marker on guitar player is different (nominative in (2), and accusative in (3)), which implies that the semantic roles should be reversed between (2) and (3), such that the translation for (2) should be as indicated, but the translation for (3) should be something like "the gifted singer discovered the talented guitar player". If so, then this needs to be revised accordingly. Unless I am missing something.

Reviewer #2: The authors thoughtfully and satisfyingly addressed the raised concerns and comments. I think that the predictions were more clearly explained and easier to follow in the response than in the manuscript, but I don’t think this necessitates changes to the manuscript.

Overall, I think this will be a very interesting study and wish the best to the authors. I’m looking forward to the future results.

7. PLOS authors have the option to publish the peer review history of their article (what does this mean?). If published, this will include your full peer review and any attached files.

Reviewer #1: **Yes: **William Matchin

Reviewer #2: **Yes: **Torsten Kai Jachmann

---

## [Author Response · Author response to Decision Letter 1]

27 Jun 2022

We would like to express our gratitude to the editor and reviewers for the constructive feedback that has helped improve the manuscript. All changes made are highlighted yellow. Responses to each individual point are provided below. We have also changed the timeline details to match the expected start date of our experiments (p26).

Reviewer 1

I appreciate the revisions and think it essentially looks good, but I still am confused about sentence examples (2) and (3) on page 5.

The provided English translations are identical for these two sentences ~"the guitar player discovered the singer", implying that the semantic roles are identical in the two sentences. But, as I understand it, the case marker on guitar player is different (nominative in (2), and accusative in (3)), which implies that the semantic roles should be reversed between (2) and (3), such that the translation for (2) should be as indicated, but the translation for (3) should be something like "the gifted singer discovered the talented guitar player". If so, then this needs to be revised accordingly. Unless I am missing something.

Thank you for pointing this out, we have revised the translation of (3) in p5.

Reviewer 2

The authors thoughtfully and satisfyingly addressed the raised concerns and comments. I think that the predictions were more clearly explained and easier to follow in the response than in the manuscript, but I don’t think this necessitates changes to the manuscript.

Overall, I think this will be a very interesting study and wish the best to the authors. I’m looking forward to the future results.

Thank you for the feedback, we have added a part of the discussion from the previous response letter into the predictions to improve the clarity of the subsection (p25).

---

## [Editor Report · Decision Letter 2]

15 Jul 2022

Neural Correlates of Thematic Role Assignment for Passives in Standard Indonesian

PONE-D-22-01301R2

Dear Dr. Jap,

We’re pleased to inform you that your manuscript has been judged scientifically suitable for publication and will be formally accepted for publication once it meets all outstanding technical requirements.

Kind regards,

Daniel Mirman

Academic Editor

PLOS ONE
---

## [Editor Report · Acceptance letter]

25 Jul 2022

PONE-D-22-01301R2 

Neural correlates of thematic role assignment for passives in Standard Indonesian 

Dear Dr. Jap:

I'm pleased to inform you that your manuscript has been deemed suitable for publication in PLOS ONE. Congratulations! Your manuscript is now with our production department. 

Kind regards, 

on behalf of

Dr. Daniel Mirman 

Academic Editor

PLOS ONE